# Effect of the Number of Anchoring and Electron-Donating Groups on the Efficiency of Free-Base- and Zn-Porphyrin-Sensitized Solar Cells

**DOI:** 10.3390/ma12040650

**Published:** 2019-02-21

**Authors:** Raheleh Nasrollahi, Luis Martín-Gomis, Fernando Fernández-Lázaro, Saeed Zakavi, Ángela Sastre-Santos

**Affiliations:** 1Área de Química Orgánica, Instituto de Bioingeniería, Universidad Miguel Hernández, Avda. de la Universidad s/n, 03203 Elche, Spain; rnasrolahi@iasbs.ac.ir (R.N.); luis.martin@umh.es (L.M.-G.); fdofdez@umh.es (F.F.-L.); 2Department of Chemistry, Institute for Advanced Studies in Basic Sciences (IASBS), Zanjan 45137-66731, Iran; zakavi@iasbs.ac.ir

**Keywords:** porphyrins, DSSC, multiple anchoring group, electron donating groups

## Abstract

A series of porphyrin compounds, free base (H_2_P) and their Zn (II) metallated analogues (ZnP), bearing one, two or three carboxylic acid groups, have been synthesized, characterized, and used as sensitizers in dye sensitized solar cells (DSSCs). The performance of these devices has been analyzed, showing higher efficiencies of those sensitized with ZnP compounds. These results have been explained, on one hand, taking into account the electronic character of the metal ion, which acts as mediator in the injection step, and, on the other, considering the number of anchoring groups, which determines both the stereoelectronic character of the dye and the way it binds to TiO_2_ surface.

## 1. Introduction

Porphyrins [1] have been extensively used as key components in different organic optoelectronic applications. Thanks to their aromatic nature, porphyrins present interesting light absorption/emission and electroactive properties, easily tunable due to their chemical versatility [2,3]. With these elements, it is possible to find in the literature a myriad of different porphyrin compounds, chemically designed to fulfill technological requirements in a variety of fields [4,5]. Dye sensitized solar cells (DSSC) [6,7,8] is one of these fields where porphyrin compounds have been widely employed, usually playing the role of sensitizing dyes. After thorough investigations, there is a general agreement on the structural requirements of porphyrins to be used in DSSCs: 1) at least one anchoring group must be present for the covalent binding to the semiconductor surface [9], 2) metallic complexes (MP), especially the zinc ones, are preferred to free-base (H_2_P), because of their longer-lived singlet excited states and much lower oxidation potentials [10], and 3) bulky electron-donor meso-substituents favor electron injection in the semiconductor, as they originate an intrinsic dipole moment [11]. High efficiencies have been achieved for TiO_2_-based devices sensitized, for example, with free-base [12] and zinc porphyrin derivatives [13], presenting one or more carboxylic acid appends as anchoring groups, either in β [9,14] or meso [15,16] positions of the porphyrin central core, and also with multiple donor groups at the meso positions [17,18,19]. The combination of the donor groups, and the electron-withdrawing carboxy group, contributes to create the push–pull effect, channelling the photoexcited electrons toward TiO_2_ and improving charge separation.

Till now few examples analyzing the photovoltaic performance of DSSC devices sensitized with porphyrin dyes as a function of the number of anchoring groups have been reported [20,21,22,23]. Here we present the synthesis and characterization of a series of free-base porphyrins with one, two, and three carboxy groups, H_2_P-CO_2_H **1**, cis-H_2_P-(CO_2_H)_2_
**2-c**, H_2_P-(CO_2_H)_3_
**3**, and their Zn metalated analogues ZnP-CO_2_H **4**, cis-ZnP-(CO_2_H)_2_
**5-c** and ZnP-(CO_2_H)_3_
**6** (Figure 1). Also, the number of electron donating –OCH_3_ groups, decreased in the series from nine (in **1** and **4**) to three (in **3** and **6**), and this is expected to tune the energy of the porphyrin frontier orbitals, influencing the π resonance interactions between porphyrin and aryl group π systems. All of them were then incorporated in efficient dye sensitized solar devices, comparing the performance obtained in terms of the number of anchoring carboxy and electron-donating -OCH_3_ groups, and the presence of zinc as metallic central ion. 

## 2. Materials and Methods 

### 2.1. Synthesis and Characterization of New Compounds

All chemicals were reagent grade, purchased from commercial sources, and were used as received unless otherwise specified. Column chromatography was performed on SiO_2_ (40–63 μm). TLC plates coated with SiO_2_ 60F254 were visualized under UV light. NMR spectra were acquired on a Bruker AC 300 spectrometer (Bruker, Billerica, MA, USA). UV/Vis spectra were recorded on a Helios Gamma spectrophotometer. Fluorescence spectra were recorded on a Perkin-Elmer LS 55 luminescence spectrometer (PerkinElmer, Waltham, MA, USA). Matrix-assisted laser desorption/ionization time-of-flight (MALDI-TOF) mass spectra were obtained on a Bruker Microflex spectrometer (Bruker, Billerica, MA, USA). Differential pulse voltammetry measurements were performed at 298 K in a conventional three-electrode cell using a m-AUTOLAB type III potentiostat/galvanostat (Metrohm, Herisau, Switzerland). Sample solutions (ca. 0.5 mM) were prepared in deaerated PhCN, containing 0.10 M tetrabutylammonium hexafluorophosphate (TBAPF_6_) as supporting electrolyte. A glassy carbon (GC) working electrode, an Ag/AgNO_3_ reference electrode, and a platinum wire counter electrode were used. Ferrocene/ferrocenium was the internal standard for all measurements.

#### 2.1.1. Synthesis of Free-Base Porphyrins **H_2_P-(CO_2_Me)_n_ 7–9**

In a 500 mL round bottom flask equipped with a magnetic stirrer, 3.53 g of 3,4,5-trimethoxybenzaldehyde (18 mmol,) 0.99 g of methyl 4-formylbenzoate (6 mmol) and 1.65 mL of pyrrole (24 mmol) were refluxed for 5 h in propionic acid (250 mL). After cooling at room temperature, the resulting mixture was extracted with dichloromethane several times and, the combined extracts, washed with water and aqueous 4% NaHCO_3_ solution, dried over MgSO_4_ and evaporated. The black colored crude compound was purified by silica gel column using hexane/ethylacetate eluent mixtures to get 0.396 g (7%) of **7** H_2_P-CO_2_Me, 0.437 g (8%) of a mixture of **8** H_2_P-(CO_2_H)_2_ cis and trans isomers and 0.422 g (8%) of **9** H_2_P(CO_2_H)_3_.

**5-(4-methoxycarbonylphenyl)-10,15,20-tris(3,4,5-trimethoxyphenyl)porphyrin (H_2_P-CO_2_Me 7).**^1^H NMR (CDCl_3_: 300 MHz), δ ppm: −2.78 (s, 2H, −NH), 3.97 (s, 18H, m−OCH_3_), 4.12 (s, 3H, −OCH_3_) 4.19 (s, 9H, p−OCH_3_), 7.47 (s, 6H, phenyl H), 8.32 (d, 2H, phenyl H), 8.44 (d, 2H, phenyl H), 8.81 (d, 2H, pyrrole H), 8.98 (d, 6H, pyrrole H). UV-vis (THF) λ_max_/nm (log Ԑ): 420 (5.64), 515 (4.36), 550 (3.98), 592 (3.74), 650 (3.00). HRMS (MALDI-TOF): m/z calcd, for C_55_H_50_N_4_O_11_ ([M^+^]): 942.35, found 942.43.

**5,10-bis(4-methoxycarbonylphenyl)-15,20-bis(3,4,5 trimethoxyphenyl)porphyrin (H_2_P-(CO_2_Me)_2_ 8, cis/trans mixture).**^1^H NMR (CDCl_3_: 300 MHz), δ ppm: −2.79 (s, 2H, −NH), 3.97 (s, 12H, m−OCH_3_), 4.11 (s, 6H, −OCH_3_) 4.18 (s, 6H, p−OCH_3_), 7.46 (s, 4H, phenyl H), 8.31 (d, 4H, phenyl H), 8.44 (d, 4H, phenyl H), 8.81 (d, 4H, pyrrole H), 8.97 (d, 4H, pyrrole H). UV-vis (THF) λ_max_/nm (log Ԑ): 421 (5.58), 515 (4.29), 549 (3.88), 591 (3.68), 649 (3.18). HRMS (MALDI-TOF): m/z calcd, for C_54_H_46_N_4_O_10_ ([M^+^]): 910.32, found 910.45. 

**5,10,15-tris(4-methoxycarbonylphenyl)-20-(3,4,5-trimethoxyphenyl)porphyrin (H_2_P-(CO_2_Me)_3_ 9).**^1^H NMR (CDCl_3_: 300 MHz), δ ppm: −2.80 (s, 2H, −NH), 3.97 (s, 6H, m−OCH_3_), 4.12 (s, 9H, −OCH_3_) 4.18 (s, 3H, p−OCH_3_), 7.46 (s, 2H, phenyl H), 8.31 (d, 2H, phenyl H), 8.43 (d, 2H, phenyl H), 8.81 (d, 2H, pyrrole H), 8.98 (d, 2H, pyrrole H). UV-vis (THF) λ_max_/nm (log Ԑ): 419 (5.44), 514 (4.13), 550 (3.73), 590 (3.48), 649 (2.30). HRMS (MALDI-TOF): m/z calcd, for C_53_H_42_N_4_O_9_ ([M^+^]): 878.30, found 878.64.

#### 2.1.2. Synthesis of Free-Base Porphyrins **H_2_P-(CO_2_H)_n_ 1–3:**

50 mg of free-base porphyrin was dissolved in a mixture of THF/MeOH (20/14 mL) and 6 mL of NaOH 20% aqueous solution were added. The crude was heated for 2 h. After cooling, the reaction crude was then diluted with dichloromethane and washed, first with HCl (1M) and then with water. The organic layer dried over MgSO_4_ and evaporated. The residue was recrystallized in hexane to get the pure powder. 

Column chromatography was conducted for H_2_P-(CO_2_H)_2_
**2**, to isolate cis and trans isomers (SiO_2_, chloroform/acetone mixtures as eluents).

**5-(4-carboxyphenyl)-10,15,20-tris(3,4,5-trimethoxyphenyl)porphyrin (H_2_P-CO_2_H 1).** Yield: 95%. ^1^H NMR (DMSO: 300 MHz), δ ppm: −2.93 (s, 2H, −NH), 3.90 (s, 18H, m−OCH_3_), 4.00 (s, 9H, p−OCH_3_), 7.53 (s, 2H, phenyl H), 8.33 (d, 2H, phenyl H), 8.39 (d, 2H, phenyl H), 8.81 (d, 2H, pyrrole H), 8.96 (d, 6H, pyrrole H). UV-vis (THF) λ_max_/nm (log Ԑ): 420(5.38), 515(4.13), 550(3.70), 592(3.70), 650 (3.30). HRMS (MALDI-TOF) calcd for C_54_H_48_N_4_O_11_ ([M^+^]): 928.33, found 928.38.

**5,10-bis(4-carboxyphenyl)-15,20-bis(3,4,5 trimethoxyphenyl)porphyrin, trans isomer (trans-H_2_P-(CO_2_H)_2_ 2-t).** Yield: 23%. ^1^H NMR (DMSO: 300 MHz), δ ppm: −2.93 (s, 2H, −NH), 3.89 (s, 12H, m−OCH_3_), 3.99 (s, 6H, p−OCH_3_), 7.54 (s, 4H, phenyl H), 8.33 (d, 4H, phenyl H), 8.38 (d, 4H, phenyl H), 8.82 (d, 4H, pyrrole H), 8.98 (d, 4H, pyrrole H). UV-vis (THF) λ_max_/nm (log Ԑ): 420 (5.59), 515 (4.27), 550 (3.93), 592 (3.78), 651(3.65). HRMS (MALDI-TOF): m/z calcd, for C_52_H_42_N_4_O_10_ ([M^+^]): 882.29, found 882.518.

**5,10-bis(4-carboxyphenyl)-15,20-bis(3,4,5 trimethoxyphenyl)porphyrin, cis isomer (cis-H_2_P-(CO_2_H)_2_ 2-c).** Yield: 71%. ^1^H NMR (DMSO: 300 MHz), δ ppm: −2.93 (s, 2H, −NH), 3.90 (s, 12H, m−OCH_3_), 3.99 (s, 6H, p−OCH_3_), 7.53 (s, 4H, phenyl H), 8.32 (d, 4H, phenyl H), 8.38 (d, 4H, phenyl H), 8.83 (d, 4H, pyrrole H), 8.98 (d, 4H, pyrrole H). UV-vis (THF) λ_max_/nm (log Ԑ): 420 (5.59), 515 (4.27), 550 (3.93), 591 (3.78), 649 (3.65). HRMS (MALDI-TOF): m/z calcd, for C_52_H_42_N_4_O_10_ ([M^+^]): 882.29, found 882.52.

**5,10,15-tris(4-carboxyphenyl)-20-(3,4,5-trimethoxyphenyl)porphyrin (H_2_P-(CO_2_H)_3_ 3).** Yield: 94%. ^1^H NMR (DMSO: 300 MHz), δ ppm: −2.93 (s, 2H, −NH), 3.90 (s, 6H, m−OCH_3_), 3.99 (s, 3H, p−OCH_3_), 7.55 (s, 2H, phenyl H), 8.35 (d, 6H, phenyl H), 8.38 (d, 6H, phenyl H), 8.85 (d, 6H, pyrrole H), 8.98 (d, 2H, pyrrole H). UV-vis (THF) λ_max_/nm (log Ԑ): 419(5.52), 515(4.18), 549(3.78), 590(3.54), 649 (3.40). HRMS (MALDI-TOF): m/z calcd, for C_50_H_36_N_4_O_9_ ([M^+^]): 836.25, found 836.50.

#### 2.1.3. Synthesis of Zinc Porphyrins **ZnP-(CO_2_H)_n_ 4–6**

Free-base porphyrin (**7**, **8** or **9**, 50 mg) and zinc acetate (1:5 mol ratio) were refluxed in dichloromethane and methanol (1:1 ratio, 40 mL each) until free-base porphyrin was completely metalated, checked by TLC and UV-vis absorption spectroscopy. The reaction mixture was then diluted with dichloromethane and washed, first with HCl (1 M) and then with water. The organic layer was collected and, after evaporation of solvent, the crude compound was purified by silica gel column using hexane/ethyl acetate eluent mixtures. Quantitative yields were obtained in all cases and isolated compounds were hydrolized following the same procedure used for free-base porphyrins hydrolisis (see Section 2.1.2).

Column chromatography was conducted for ZnP-(CO_2_H)_2_
**5** to isolate cis and trans isomers (SiO_2_, chloroform/acetone mixtures as eluents).

**Zinc(II) 5-(4-carboxyphenyl)-10,15,20-tris(3,4,5-trimethoxyphenyl)porphyrinate (ZnP-CO_2_H 4**). Yield: 98%. ^1^H NMR (DMSO: 300 MHz), δ ppm: 3.90 (s, 18H, m−OCH_3_), 3.99 (s, 9H, p−OCH_3_), 7.45 (s, 2H, phenyl H), 8.29 (d, 2H, phenyl H), 8.37 (d, 2H, phenyl H), 8.75 (d, 2H, pyrrole H), 8.90 (d, 6H, pyrrole H). UV-vis (THF) λ_max_/nm (log Ԑ): 426 (5.63), 557 (4.20), 597 (3.65). HRMS (MALDI-TOF) calcd for C_54_H_46_N_4_O_11_ ([M^+^]): 990.25, found 989.22.

**Zinc(II) 5,10-bis(4-carboxyphenyl)-15,20-bis(3,4,5-trimethoxyphenyl)porphyrinate, trans isomer (trans-ZnP-(CO_2_H)_2_ 5-t).** Yield: 20%. ^1^H NMR (DMSO: 300 MHz), δ ppm: 3.89 (s, 12H, m−OCH_3_), 3.98 (s, 6H, p−OCH_3_), 7.46 (s, 4H, phenyl H), 8.23 (d, 4H, phenyl H), 8.35 (d, 4H, phenyl H), 8.76 (d, 4H, pyrrole H), 8.91 (d, 4H, pyrrole H). HRMS (MALDI-TOF): m/z calcd, for C_52_H_40_N_4_O_10_Zn ([M^+^]): 944.20, found 943.45.

**Zinc (II) 5,10-bis(4-methoxycarbonylphenyl)-15,20-bis(3,4,5-trimethoxyphenyl)porphyrinate, cis isomer (cis-ZnP-(CO_2_H)_2_ 5-c).** Yield: 69%. ^1^H NMR (DMSO: 300 MHz), δ ppm: 3.89 (s, 12H, m−OCH_3_), 3.98 (s, 6H, p−OCH_3_), 7.45 (s, 4H, phenyl H), 8.23 (d, 4H, phenyl H), 8.34 (d, 4H, phenyl H), 8.76 (d, 4H, pyrrole H), 8.91 (d, 4H, pyrrole H). UV-vis (THF) λ_max_/nm (log Ԑ): 426 (5.56), 557 (4.29), 597 (3.80). HRMS (MALDI-TOF): m/z calcd, for C_52_H_40_N_4_O_10_Zn ([M^+^]): 944.20, found 943.48.

**Zinc(II) 5,10,15-tris(4-carboxyphenyl)-20-(3,4,5-trimethoxyphenyl)porphyrinate (ZnP-(CO_2_H)_3_ 6).** Yield: 98%. ^1^H NMR (DMSO: 300 MHz), δ ppm: 3.90 (s, 6H, m−OCH_3_), 3.99 (s, 3H, p−OCH_3_), 7.47 (s, 2H, phenyl H), 8.29 (d, 6H, phenyl H), 8.37 (d, 6H, phenyl H), 8.78 (d, 6H, pyrrole H), 8.94 (d, 2H, pyrrole H). UV-vis (THF) λ_max_/nm (log Ԑ): 426 (5.70), 557 (4.33), 598 (3.89). HRMS (MALDI-TOF): m/z calcd, for C_50_H_34_N_4_O_9_Zn ([M^+^]): 898.16, found 897.40.

### 2.2. Device Preparation

Double-layered nanoporous TiO_2_ photoanodes were prepared coating pastes of anatase TiO_2_ nanoparticles having two different diameters, 20 nm (Dyesol’s 90 T) and 400 nm (Dyesol’s WER2-O), onto TiCl_4_ treated FTO glass plates (TEC 15 A, 2.2 mm, Xop Glass), by repetitive screen printing to obtain the required thickness. These electrodes were gradually heated under a programmed flow: at 370 °C for 10 min and 450 °C for 10 min. Their apparent surface area was 0.16 cm^2^ (0.4 cm × 0.4 cm), and revealed a total thickness of 8–10 μm, containing a 3–4 μm scattering layer. The TiO_2_ electrodes were treated again with TiCl_4_ under 70 °C for 30 min and sintered at 500 °C for 30 min, before they were dipped into dye solution. The nanocrystalline TiO_2_ films were immersed into 5 mM dye solutions, without any other additives, i.e. co-adsorbents, and kept at RT for 20 h. Finally, dye adsorbed TiO_2_ photoanodes and thermally platinized and drilled FTO counter electrodes (TEC 8 A, 3 mm, Xop Glass), were assembled into sandwich type cells, separated by a 30 μm thick hot-melt gasket (Surlyn, Dupont), and sealed by heating. An electrolyte solution (0.1 M LiI, 0.03 M I_2_, 0.5 M 4-*tert*-butylpyridine, 0.1 M guanidinium thiocyanate, 1 M 1-butyl-3-methylimidazolium iodide in acetonitrile/valeronitrile 85:15 v/v) was introduced in the assembled devices. A series of three devices with each dye were prepared and their photovoltaic performance measured. The values described are, in all cases, the best obtained, with no significant differences between devices sensitized with the same dye, which ensures the reproducibility and consistency of the results.

### 2.3. Photovoltaic Characterization

An ABET 150W xenon light source equipped with an AM 1.5 G correcting filter was employed. The light intensity was adjusted to 100 mW/cm^2^ (the equivalent of 1 sun), prior to every measurement, using a calibrated photovoltaic reference cell (15150, ABET Technologies). The applied potential and cell current were registered with a Keithley 2401 low voltage digital sourcemeter. The incident photon-to-current conversion efficiency (IPCE) was measured as a function of wavelength from 400 to 800 nm by using an IPCE-DC system (Lasing SA).

## 3. Results and Discussion

### 3.1. Synthesis of New Compounds

Free base porphyrin compounds H_2_P-CO_2_H **1**, H_2_P-(CO_2_H)_2_
**2** (cis and trans) and H_2_P-(CO_2_H)_3_
**3** were prepared through a two-step synthetic sequence, as it is described in Scheme 1. First, methyl ester derivatives (**7**–**9**) were obtained as pure compounds, following the traditional Adler-Longo method [24], by reacting a 4:3:1 mixture of pyrrole, 3,4,5-trimethoxybenzaldehyde and methyl 4-formylbenzoate in propionic acid. In a second step, methyl ester groups were hydrolyzed by heating in an aqueous base solution affording, in almost quantitative yields, free-base porphyrins with one, two and three carboxylic groups (**1**–**3**). It is worth to note that the hydrolysis of H_2_P-(CO_2_Me)_2_
**8**, mixture of cis and trans stereoisomers, gave a new mixture which could be resolved into its components, cis-H_2_P-(CO_2_H)_2_
**2-c** and trans-H_2_P-(CO_2_H)_2_
**2-t**, through standard column chromatography.

Finally, in order to obtain Zn porphyrins ZnP-CO_2_H **4**, ZnP-(CO_2_H)_2_
**5** (cis and trans) and ZnP-(CO_2_H)_3_
**6**, methyl ester precursors (**7**–**9**) were, first metallated in refluxing dichloromethane/methanol mixture, in presence of a zinc acetate excess, and, without isolation, hydrolyzed, following the procedure previously used in the synthesis of H_2_P-(CO_2_H)_n_
**1**–**3**. An example of the synthetic sequence performed (synthesis of ZnP-CO_2_H **4**) is shown in Scheme 2. As it occurred with H_2_P-(CO_2_H)_2_
**2** cis and trans isomers, the metalation of **8** (mixture of isomers), followed by basic hydrolysis, afforded a new mixture of compounds which could be separated into its components, ZnP-(CO_2_H)_2_
**5** cis and trans, through standard column chromatography.

All synthesized compounds **1**–**9** (Appendix A were characterized through common techniques, such ^1^H NMR and UV-vis spectroscopies and HR-MS (MALDI TOF) mass spectrometry. In this context, the ^1^H NMR signals obtained for H_2_P-(CO_2_H)_n_
**1**–**3** (Appendix A), ZnP-(CO_2_H)_n_
**4**–**6** (Appendix A) and H_2_P-(CO_2_Me)_n_
**7**–**9** (Appendix A) showed similar chemical displacements, but displayed different integral values for the signals corresponding to the phenyl and methoxy groups. On the other hand, HR MS gave, in all cases, a single peak with a m/z ratio that exactly matched the calculated one (Appendix A).

### 3.2. Optical and Electrochemical Properties

Free porphyrins **1**, **2** cis, and **3**, and zinc derivatives, **4**, **5** cis, and **6**, were evaluated as sensitizers for DSSC devices. It must be stated that trans isomers of compounds **2** and **5** could not be evaluated, because of their reduced solubility which prevented their optical and electrochemical characterisation.

The UV-vis absorption spectra of all dyes show typical features of porphyrin compounds (Figure 2). While H_2_P **1**–**3** present a strong absorption band at 420 nm (S_0_→S_2_, Soret band) and four weak transitions to the first excited state between 500 and 680 nm (S_0_→S_1_, Q bands), ZnP **4**–**6** exhibit, a 6–7 nm red-shifted Soret band and only two Q bands, which are located in the 550–650 nm area. These spectral changes, upon metalation of the macrocycle, are probably due to the increased symmetry of the porphyrin core, moving from D_2h_ to D_4h_. It is also worth to note that the introduction of a zinc atom in the porphyrin cavity, causes a π-π interaction between the metal p_π_ orbital and the porphyrin π system. According to the four orbital model of porphyrins [25], the electronic density on the meso positions and the pyrrolic nitrogen atoms is large, so the energy of a_2u_ and e_g_ orbitals is, therefore, influenced by both the metal ion and the substituents introduced at the meso positions. Regarding to the molar extinction coefficients, zinc derivatives show somehow higher values than the metal free ones. It is also remarkable that within a series (ZnP/H_2_P), variations in the coefficients are minimal, and are attributed to differences in solubility, due to number of carboxy groups. Finally, internal conversion between S_2_ and S_1_ is rapid, so fluorescence is only detected from S_1_. Taking into account that intensity of Q bands is weak, transition energies cannot be accurately estimated from the intersection of normalized absorption and emission spectra [26,27], so the optical band gap (E_g_^opt^) was here calculated using the equation (1) where λ_edge_ is the onset value of the absorption spectrum in the direction of longer wavelengths [28].
(1)$$E_{g}^{\mathrm{opt}}=\frac{1240}{\lambda_{\mathrm{edge}}}\mathrm{eV}$$

Peak position (λ_abs_), molar absorption coefficients (ε) of Soret and Q bands, onset values (λ_edge_) and estimated optical band gap (E_g_^opt^) of porphyrin dyes are listed in Table 1.

Electrochemical measurements were performed for H_2_P **1**–**3** and ZnP **4**–**6**, registering differential pulse voltammograms in benzonitrile solution (Figure 3). All measured dyes exhibit simple, clear and sharp waves in the anodic part, very different from those in the cathodic area. This can be probably due to the genuine electron-donor character of porphyrin compounds and, particularly, for these studied 3,4,5-trimethoxyphenyl–substituted porphyrins. Also, this observation provides evidence for extensive changes in the electronic structure of the studied compounds, induced by one electron reduction of the aromatic macrocycles. In this context, and taking into account only oxidation processes, H_2_P **1**–**3** are more resistant to oxidation than ZnP **4**–**6**. On the other hand, as a general tendency, oxidation potentials increase with the number of carboxy groups. Due to the increase in the number of carboxy groups, which is associated with a concomitant decrease in the number of trimethoxyphenyl moieties, the resonance interactions between the a_2u_ orbital and the porphyrin π system become weaker, thus favoring the stabilization of that orbital. The decreased oxidation potential of the metalloporphyrins, compared to that of the free base analogues, seems to be due to destabilization of a_2u_ orbitals of the former, probably caused by π resonance interactions between the metal p_π_ orbital, porphyrin a_2u_ orbital and the aryl group π system [29].

Spectral and electrochemical properties allow to determine the electron injection and dye regeneration possibility during DSSC performance. From E_ox_ potential values, referenced to Fc/Fc^+^ pair, (Table 2) HOMO energy level for all dyes can be easily calculated, using the Equation (2).
(2)$$E_{\mathrm{HOMO}}\left(\mathrm{eV}\right)=-4.8-E_{\mathrm{ox}}{(V\;\mathrm{vs}\;\mathrm{Fc}/\mathrm{Fc}}^{+})$$

These values, combined with the previously estimated optical band gap (E_g_^opt^), and values of conduction band of TiO_2_ (−4.2 eV) [30] and I^−^/I_3_^−^ redox potential (−4.89 eV) [31,32], allow to sketch an energy diagram representing HOMO and LUMO levels for all studied dyes (Figure 4). At this point, it is important to mention that the determination of the LUMO level, using the E_HOMO_ obtained from electrochemical measurements and estimated E_g_^opt^, is a very useful approximation in the case that both E_ox_ and E_red_ values cannot be accurately extracted from electrochemical measurements. As can be seen, LUMO energy levels are, in all cases, higher than TiO_2_ conduction band (TiO_2_ CB), fundamental requisite to make the electron injection thermodynamically feasible, while HOMO levels are, always, lower than I^−^/I_3_^−^ redox potential, making possible the regeneration of the oxidized dye.

### 3.3. Preparation of Devices and Photovoltaic Characterization

Dye sensitized solar cells were prepared following a standard procedure, using double layered screen-printed TiO_2_ photoanodes, platinum casted counter-electrodes, and liquid electrolyte containing 0.05 M LiI, 0.03 M I_2_, 0.5 M 4-*tert*butylpyridine (TBP), 0.1 M guanidinium thiocyanate (GNCS), 1 M 1-butyl-3-methylimidazolium iodide (BMII) in acetonitrile/valeronitrile (85:15 v/v). Once prepared, efficiencies of all devices were evaluated under standard Air Mass 1.5 global (AM1.5G) solar irradiation, constructing J/V curves, and analyzing incident photon to current conversion efficiencies (IPCE).

All employed dyes gave efficient photoanode sensitization without any other additive (20 h dipping in 5 mM dye ethanol solution), qualitatively appreciated through a deep anode coloration. At this point we considered the use of co-adsorbents to improve the performance, particularly chenodeoxycolic acid (Cheno). In porphyrin and phthalocyanine sensitized solar cells, Cheno is commonly used as co-adsorbent and, directly incorporated in sensitizing solutions, improves both J_sc_ and V_oc_ parameters, thus leading to better device efficiencies, so we prepared sensitizing solutions of our dyes, incorporating Cheno as co-adsorbent. Unfortunately, only scarce sensitization occurred in all cases, obtaining, after dipping, soft-colored photoanodes unable to be efficiently photoexcitated. In this case, an unbalanced competency between dye and Cheno molecules seems to occur, hampering the anchorage of sensitizing units onto TiO_2_ surface.

Figure 5 shows J/V curves for devices sensitized with H_2_P and ZnP derivatives, and Table 3 resumes the photovoltaic parameters, short-circuit current (J_sc_), open circuit voltage (V_oc_) and fill factor (FF), reflecting much better efficiencies for the zinc compounds. The reason for this difference must be found in the Zn^+2^ porphyrin metallic core, a closed shell ion with empty coordination sites, which allow an efficient and rapid injection of photoexcited porphyrin electrons to TiO_2_ conduction band, acting as mediator for electron transfer from I^−^/I_3_^−^ to the a_2u_ orbital of the porphyrinsensitizer. It is worth to note that the obtained efficiency for device sensitized with ZnP-CO_2_H **4** is up to 1.62%, with J_sc_ value of 4.34 mA/cm^2^, V_oc_ of 0.57 V and FF of 0.65, better than that previously reported for the same compound, (1.06%) [33], demonstrating the convenience of our device preparation protocol. On the other hand, ZnP derivatives with two (cis-ZnP-(CO_2_H)_2_
**5-c**) and three (ZnP-(CO_2_H)_3_
**6**) carboxylic acid anchoring groups showed lower results, due a combination of both electronic and structural features. It is well known that efficient dyes usually present the so called push–pull effect, thus facilitating the injection to the TiO_2_ conduction band [34]. ZnP-CO_2_H **4** shows strong push pull directionality, thanks to the already mentioned electron-acceptor character of the anchoring group and the presence of three electron-donor trimethoxyphenyl substituents in the meso positions. A lesser push pull effect can be found in cis-ZnP-(CO_2_H)_2_
**5-c** and ZnP-(CO_2_H)_3_
**6**. On the other hand, more than one anchoring group means a better anchorage to TiO_2_, but not necessarily means better performance. Anchored dyes in such way could adopt a flat binding position (“face-to-face”, Figure 6a) referred to the TiO_2_ surface, offering a lower coverage degree than it does in a perpendicular/vertical fashion (“edge-to-face”, Figure 6b) [35,36]. This is what probably happens, looking to the photovoltaic parameters extracted from J/V curves. The J_sc_ value for ZnP-CO_2_H **4** sensitized devices, compared to those sensitized with cis-ZnP-(CO_2_H)_2_
**5-c** and ZnP-(CO_2_H)_3_
**6** (Table 3), indicates a higher photogenerated current, due to a more compact coverage of the semiconductor surface by the dye. In this context, high values of V_oc_ in ZnP-CO_2_H **4** sensitized devices (Table 3), also confirm this hypothesis. High V_oc_ values are indicative of non-aggregated molecules and, in this case, also protection of the central metal core. If ZnP-CO_2_H **4** molecules are covering the TiO_2_ surface in an “edge-to-face” manner, π-π stacking phenomena seems unlikely to happen between adjacent molecules, due to the necessary non-coplanar position (related to the porphyrin flat structure) of bulky meso-trimetoxyphenyl groups. This results in an effective protection of the porphyrin central metal core, avoiding early recombination processes with the electrolyte, thus affording better V_oc_ values. Same reasoning could be applied to devices sensitized with H_2_P derivatives **1**–**3** but, in this case, the lack of metal ion in the dye becomes crucial. In absence of such mediator, the injection step is slowed down, and early recombination processes are then favored. Interestingly, and opposite to what happens with ZnP **4**–**6** sensitized devices, structural features of free-base dyes seem to gain weight *vs* electronic characteristics (more than one binding group *vs* push-pull effect). cis-H_2_P-(CO_2_H)_2_
**2-c** and H_2_P-(CO_2_H)_3_
**3** sensitized devices show better J_sc_ values than those of H_2_P-(CO_2_H) **1** (Table 3). This fact indicates that the chromophore is closer to the semiconductor surface, balancing out the absence of metal ion. Furthermore, the number of sterically demanding trimethoxyphenyl groups of the dye, is also reflected in V_oc_ values, higher in the case of cis-H_2_P-(CO_2_H)_2_
**2-c** sensitized devices. The presence of two bulky trimethoxyphenyl groups at the meso positions, and only one in the case of H_2_P-(CO_2_H)_3_
**3**, partially avoids π-π stacking phenomena between adjacent adsorbed molecules.

Finally, IPCE measurements were made for all devices, showing maxima of photogenerated current in wavelengths which match the absorption profile of employed sensitizers (Figure 7). As expected, higher performances were obtained for ZnP derivatives **4**–**6**, with maxima at 420, 560 and 600 nm and percentages of 50%, 14%, and 9% respectively in the case of ZnP-CO_2_H **4**. These results confirm the convenience of introducing just one anchoring place in carboxy ZnP-based sensitizers for DSSCs.

## 4. Conclusions

A series of unsymmetric porphyrin compounds, free-base [H_2_P-(CO_2_H)_n_
**1**–**3**] and Zn metallated [ZnP-(CO_2_H)_n_
**4**–**6**], with one, two or three carboxyphenyl anchoring groups, were synthesized, characterized and used as sensitizers in TiO_2_-based DSSC devices. The comparison of their performances shows the utility of these compounds for this use, reflecting that ZnP-(CO_2_H)_n_
**4**–**6** sensitized solar cells offer better efficiencies, compared to those sensitized with H_2_P-(CO_2_H)_n_
**1**–**3**. This is due to the presence of a zinc ion in the porphyrin inner cavity, acting as electronic mediator in the injection step. The observed order of efficiency for the zinc complexes, i.e. ZnP-CO_2_H **4** > ZnP-(CO_2_H)_3_
**6** > cis-ZnP-(CO_2_H)_2_, **5-c** is in agreement with a rapid injection of the photoexcited electrons to the TiO_2_ conduction band, where electronic characteristics of the dye prevail over its structural features. On the other hand, the observed order of efficiency for free-base dyes, i.e., cis-H_2_P-(CO_2_H)_2_
**2-c** > (H_2_P-(CO_2_H)_3_
**3** > H_2_P-CO_2_H **1**, agrees with the absence of a metallic mediator, slowing down the injection step and making structural features of the dye to gain prominence over its electronic characteristics.

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
