# Peer review of "Effect of the Number of Anchoring and Electron-Donating Groups on the Efficiency of Free-Base- and Zn-Porphyrin-Sensitized Solar Cells"

_materials, 2019, doi:10.3390/ma12040650_

Round 1
Reviewer 1 Report
The authors report on the synthesis and characterization of a set of porphyrin derivatives. Their implementation in DSSCs is also investigated. The manuscript is in principle of interest to thee readres of Materials. However, there are a number of issues that need to be addressed.
- The references must be updated. A big chunk of them is older than 10 years, and this is not acceptable for a fast growing field of research such as unconventional photovoltaics.
-How many devices of the same type have been prepared? Are the values of the cell's figures of merit average or maximum? Which is the spread? This information is relevant, in view of assessing the reliability of the devices, and must be included into the main text.
-The trend of the photovoltaic parameters by varying the porphyrin structures should be better highlighted and discussed, to allow the reader to gain useful insights into the dye structure-properties-device response. (At least) Tentative explanations re. the effect of the number of -CO2H groups and the presence of Zn metal on the device Jsc, Voc, FF and efficiencies should be comparatively given. This should be also recalled in the conclusion section.
-Some English editing is required
Author Response
Reviewer 1:
The authors report on the synthesis and characterization of a set of porphyrin derivatives. Their implementation in DSSCs is also investigated. The manuscript is in principle of interest to the readers of Materials. However, there are a number of issues that need to be addressed.
- The references must be updated. A big chunk of them is older than 10 years, and this is not acceptable for a fast growing field of research such as unconventional photovoltaics.
Response: We appreciate the reviewer’s valuable comments. References are now updated.
-How many devices of the same type have been prepared? Are the values of the cell's figures of merit average or maximum? Which is the spread? This information is relevant, in view of assessing the reliability of the devices, and must be included into the main text.
Response: A series of three devices with each dye were prepared and evaluated, and the values described in the manuscript are, in all cases, the best obtained. As the referee suggests, this information is relevant and now is included in the text, pointing out that no significant differences were observed, ensuring the reproducibility and consistency of the results.
The trend of the photovoltaic parameters by varying the porphyrin structures should be better highlighted and discussed, to allow the reader to gain useful insights into the dye structure-properties-device response. (At least) Tentative explanations re. the effect of the number of -CO2H groups and the presence of Zn metal on the device Jsc, Voc, FF and efficiencies should be comparatively given. This should be also recalled in the conclusion section.Response: As the referee suggested, the trend in photovoltaic parameters have now been discussed, highlighting the influence of the number of CO2H groups and presence of Zn metal ion in the efficiency showed by the cell. Abstract and conclusions sections have also been modified.
Reviewer 2 Report
This manuscript describes the effect of the number and position of donor/acceptor groups in porphyrin compounds, on sensitized solar cells. In particular, the authors present the synthesis and characterization of a series of free-base and Zn metaled meso-trimethoxyphenyl porphyrins with one, two and three carboxylic acid anchors. The authors conclude that metaled porphyrins lead to better efficiencies compared to those free-base due to the electronic nature of metallic central core, and that the presence of only one carboxylic acid group seems to be the optimal.
Despite the efficiency values are low, in general, the paper shows to have no very serious weakness. However, some minor issues should still be clarified prior to publication:
1) It would be helpful if the authors could better describe the effect of the metaled porphyrins compared to those free-base and explain the statement “due to the electronic nature of metallic central core” in the Conclusions. In addition, in the manuscript it is not clear the reason the authors claim that in the case of Zn-porphyrins the electronic effects dominate the steric effects of the meso-substituents, while in the case of free-base porphyrins the steric hindrance is dominant. In other words, why the presence of three electron-donor trimethoxyphenyl substituents in the meso positions, leading to a strong push-pull effect, is advantageous to metaled porphyrins but this is not in the case of free-base?
2) I would suggest the authors point out that the determination of LUMO level using the EHOMO obtained from electrochemical measurements and the optical band gap is an approximation, because the optical band gap does not match the electrochemical band gap.
3) The authors comment on the performances of devices only in terms of power conversion efficiency but the other photovoltaic parameters should be taken into account. Observing the variation of Voc or Jsc and FF could lead to a more comprehensive discussion.
Author Response
Reviewer 2: This manuscript describes the effect of the number and position of donor/acceptor groups in porphyrin compounds, on sensitized solar cells. In particular, the authors present the synthesis and characterization of a series of free-base and Zn metaled meso-trimethoxyphenyl porphyrins with one, two and three carboxylic acid anchors. The authors conclude that metaled porphyrins lead to better efficiencies compared to those free-base due to the electronic nature of metallic central core, and that the presence of only one carboxylic acid group seems to be the optimal.
Despite the efficiency values are low, in general, the paper shows to have no very serious weakness. However, some minor issues should still be clarified prior to publication:
1) It would be helpful if the authors could better describe the effect of the metaled porphyrins compared to those free-base and explain the statement “due to the electronic nature of metallic central core” in the Conclusions. In addition, in the manuscript it is not clear the reason the authors claim that in the case of Zn-porphyrins the electronic effects dominate the steric effects of the meso-substituents, while in the case of free-base porphyrins the steric hindrance is dominant. In other words, why the presence of three electron-donor trimethoxyphenyl substituents in the meso positions, leading to a strong push-pull effect, is advantageous to metaled porphyrins but this is not in the case of free-base?
Response: The discussion and conclusion parts have been completely modified, in order to attend referee’s concerns, and we think that the reasons that make Zn-porphyrins better dyes that free base, and why electronic effects dominate in the former, are now clearer.
2) I would suggest the authors point out that the determination of LUMO level using the EHOMO obtained from electrochemical measurements and the optical band gap is an approximation, because the optical band gap does not match the electrochemical band gap.
Response: This point is now stated in the text.
3) The authors comment on the performances of devices only in terms of power conversion efficiency but the other photovoltaic parameters should be taken into account. Observing the variation of Voc or Jsc and FF could lead to a more comprehensive discussion.
Response: Now the discussion includes photovoltaic parameters Jsc, Voc, and FF, as the referee suggests, and is now clearer
Reviewer 3 Report
Title of the manuscript: Effect of the Number of Anchoring and Electron Donating Groups on the efficiency of Free-Base- and Zn-Porphyrin-Sensitized Solar Cells Manuscript ID: Materials-427310 Journal name: Materials Comments to the Authors This research demonstrated the synthesis and application of a series of porphyrin compounds with and without Zn metal for DSSCs. Even though the DSSCs performance of these compounds is very low, this study will help the researcher to develop porphyrin complexes with other metals for DSSCs. Before considering for publication the following issues need to consider. 1. Abstract should be written clearly and concisely with the key results for DSSCs application. Currently, it is too general, which does not give the key conclusion of this work completely. Similarly, author should consider rewriting the conclusion part also. 2. Page#2, line 50 and 54. –OCH3 is written as electron donating and electron releasing group, which should be corrected. 3. The thickness of TiO2 layer is 8–10 mm, containing a 3–4 mm scattering layer with 30 mm thick surlyn film. Please check this one carefully. Usually, the best performed DSSCs utilize µm range TiO2 film with µm thickness surlyn. If it is correct, the reason for the use of high thickness film should be stated. 4. Some typos are available in the manuscript such as TiO2 should be corrected, 6 ZnP-(CO2H)3 should be changed to ZnP-(CO2H)3 6 (In table 3). Please check carefully all other typos in the manuscript. 5. I recommend to draw the Tauc plot of all dyes and insert them in the inset of Fig. 2, which will give a clear picture about the optical band gap of each compound. 6. Page#7, line # 245: ‘’All measured …reversible waves in the anodic part, but non reversible in the cathodic area’’. How come author concludes this reversible issue from the DPV responses where only oxidation reaction is occurred? 7. Usually, the DPV or CV of dyes is measured using Pt as standard working electrode. Why author choose glassy carbon working electrode instead of Pt. Will it induce any difference in the HOMO level of dyes? 8. I think Table 2 and Fig. 4 is analogous to each other. One of them can eliminate to avoid redundancy (probably Table 3). 9. How about the DSSCs performance of all dyes without co-adsorbent. 10. English of this manuscript must be edited by English Editing service. 11. Voc of is the energy gap between the Fermi level of TiO2 and the redox potential of electrolyte. Of course it is significantly dependent on the electron transfer kinetics (recombination, back reaction etc.). There is a clear difference in the Voc of all the dye based DSSCs, which should be explained with impedance analyses. 12. Density current in Fig. 5 should be changed to current density. 13. I think the EQE (Fig. 6b) is not matching with the Jsc of dye 4 and 6. Also, why there is high percent of EQE for all dyes bellow 400 nm. From the UV-Visible spectra is clear that there is almost no light absorption bellow 400 nm
Author Response
Reviewer 3: This research demonstrated the synthesis and application of a series of porphyrin compounds with and without Zn metal for DSSCs. Even though the DSSCs performance of these compounds is very low, this study will help the researcher to develop porphyrin complexes with other metals for DSSCs. Before considering for publication the following issues need to consider.
1. Abstract should be written clearly and concisely with the key results for DSSCs application. Currently, it is too general, which does not give the key conclusion of this work completely. Similarly, author should consider rewriting the conclusion part also.
Response: Abstract and conclusions sections have been modified.
2. Page#2, line 50 and 54. –OCH3 is written as electron donating and electron releasing group, which should be corrected.
Response: It is now corrected.
3. The thickness of TiO2 layer is 8–10 mm, containing a 3–4 mm scattering layer with 30 mm thick surlyn film. Please check this one carefully. Usually, the best performed DSSCs utilize µm range TiO2 film with µm thickness surlyn. If it is correct, the reason for the use of high thickness film should be stated.
Response: It was an error, µm should appears instead of mm. It is now corrected.
4. Some typos are available in the manuscript such as TiO2 should be corrected, 6 ZnP-(CO2H)3 should be changed to ZnP-(CO2H)3 6 (In table 3). Please check carefully all other typos in the manuscript.
Response: It is now corrected.
5. I recommend to draw the Tauc plo[MGL1] t of all dyes and insert them in the inset of Fig. 2[MGL2] , which will give a clear picture about the optical band gap of each compound.
Response: As the referee suggests, Tauc plots of sensitized TiO2 photoanodes would give a clear picture about the optical band gaps. But, trying to be respectful to his/her suggestion, we do not think this analysis is necessary to be performed. A Tauc plot is one method of determining the optical band gap in semiconductors, and this is not case. As it is stated in the text, we always determine the optical and electrochemical properties of our compounds previously to be adsorbed onto the semiconductor surface, and to assure that those potential dyes fulfil the energetic requirements to make the electron injection thermodynamically feasible.
6. Page#7, line # 245: ‘’All measured …reversible waves in the anodic part, but non reversible in the cathodic area’’. How come author concludes this reversible issue from the DPV responses where only oxidation reaction is occurred?
Response: It is true that, with this data, we cannot conclude about the reversibility of waves, each of them associated to single electrochemical events, and attending to referee’s concern, we have changed the text in the manuscript. The original:
‘’All measured …reversible waves in the anodic part, but non reversible in the cathodic area’’
Has been changed to
“All measured dyes exhibit simple, clear and sharp waves in the anodic part, very different from those in the cathodic area.”
Trying to point out that oxidation processes are clear and reduction ones are not, which was the originally intended meaning for this sentence.
7. Usually, the DPV or CV of dyes is measured using Pt as standard working electrode. Why author choose glassy carbon working electrode instead of Pt. Will it induce any difference in the HOMO level of dyes?.
Response: Being true that Pt is commonly used as working electrode in DPV and CV measurements, glassy carbon is also employed for this purpose. In fact, it is preferred over Pt very often, because it is less sensitive to impurities, has a wider electrochemical window and is easily refreshed by polishing its surface. These are the reasons why we usually employ this experimental set up in most of our works.
We honestly think that no significant difference in HOMO levels will be found if measurements were repeated using Pt as working electrode. In our case, HOMO level of dyes has been calculated through equation 2, which specifically includes the term Eox referred to Fc/Fc+ (internal standard). Referring oxidation potentials to Fc/Fc+ pair has the advantage that the values obtained remain almost unchanged if different combinations of counter, working and reference electrodes are used. See “Ferrocene as an internal standard for electrochemical measurements”, Inorg. Chem., 1980, 19 (9), pp 2854–2855, DOI: 10.1021/ic50211a080.
8. I think Table 2 and Fig. 4 is analogous to each other. One of them can eliminate to avoid redundancy (probably Table 3).
Response: Table 2 now is modified, avoiding redundancy.
9. How about the DSSCs performance of all dyes without co-adsorbent.
Response: All prepared devices were sensitized without co-adsorbents, and it was explicitly stated in the original manuscript. Anyway, we have made some corrections to the main text, in “materials and methods” and “results and discussion” sections, to highlight the fact that our devices are sensitized with pure dye solutions, without co-adsorbent, just to avoid any confusion at this point.
10. English of this manuscript must be edited by English Editing service.
Response: It is now corrected
11. Voc of is the energy gap between the Fermi level of TiO2 and the redox potential of electrolyte. Of course it is significantly dependent on the electron transfer kinetics (recombination, back reaction etc.). There is a clear difference in the Voc of all the dye based DSSCs, which should be explained with impedance analyses.
Response: Impedance analysis of described devices is out of the scope of this work. These measurements are currently underway and will be included in a future work, consisting in a wider comparative study of different type of porphyrin compounds.
12. Density current in Fig. 5 should be changed to current density.
Response: It is now corrected
13. I think the EQE (Fig. 6b) is not matching with the Jsc of dye 4 and 6. Also, why there is high percent of EQE for all dyes bellow 400 nm. From the UV-Visible spectra is clear that there is almost no light absorption bellow 400 nm.
Response: UV-vis absorption spectra of TiO2 thin films exhibit strong peaks in the range of 310-350 nm, which are due to the excitation of electrons from the valence band to the conduction. See “Kite, S. V; Sathe, D.J.; Patil, S.S.; Bhosale, P.N.; Garadkar, K.M. Nanostructured TiO2 thin films by chemical bath deposition method for high photoelectrochemical performance. Mater. Res. Express 2018, 6, 026411”.
Therefore, IPCE bands in the 300-400 nm region are due to this absorption profile. In order to avoid misunderstandings at this point, IPCE spectra have been modified, and now start from 400 nm.
Round 2
Reviewer 1 Report
The authors adequately addressed the points raised by the reviewer. Therefore the manuscript is now suitable for publication in Materials Journal